# Navigating Adult Life from Emerging to Middle Adulthood: Patterns of Systemic Influences and Time Perspective in Migrants

**DOI:** 10.3390/bs14020086

**Published:** 2024-01-25

**Authors:** Teresa Maria Sgaramella, Andrea Zammitti, Paola Magnano

**Affiliations:** 1FISPPA Department, University of Padova, Via Venezia 14, 35137 Padova, Italy; 2Department of Educational Sciences, University of Catania, 95124 Catania, Italy; andrea.zammitti@unict.it; 3Department of Human and Social Sciences, Kore University,94100 Enna, Italy; paola.magnano@unikore.it

**Keywords:** emerging and middle adulthood, migration, systemic influences, time perspective

## Abstract

This study emerges at the intersection of adult development and systems theory frameworks and their contributions to understanding migration experiences and associated cultural transitions. The adult development approach enables a deep understanding of the complexities that adults experience when they move from exploring themselves and their environment in emerging adulthood to establishing their identities and roles during middle adulthood. The systems theory framework, on the other hand, provides insights into the role of social and cultural dimensions in the lives of emerging and middle-adult immigrants who have navigated diverse cultures, roles, and identities. The study highlights the patterns and dynamic interactions of diverse systems of influences and their roles in shaping the self and relational identities of thirty emerging and middle-aged adults who have experienced migration.

## 1. Introduction

Adults live in contexts and systems that are characterised by high levels of insecurity, challenges, and changes. This often requires them to reshape their lives across adulthood. Adult development is a theoretical lens that can provide insights into the qualitative experiences and diverse ways of thinking, talking, and acting of people with various backgrounds [1].

According to the literature, emerging adults, whose age ranges from approximately 18 to 29 years, experience numerous concurrent changes in all facets of their lives [2]. These include leaving home for educational or career purposes and adjusting to novel situations, and it is more important for them to achieve satisfaction with their lives. Emerging adulthood can be characterised as the age of identity explorations, instability, self-focus, feeling in-between, and facing possibilities [3]. While these five dimensions generally define this life stage, the patterns and frequency of how individual adults experience instability, uncertainty, feeling in-between, and conflicts related to identity seem to vary between diverse cultural groups and collectivistic societies [4,5].

There has recently been an increased amount of attention devoted to the period that follows early adulthood: middle adulthood [6]. The corresponding age range varies across studies, but it is widely accepted to be between 30 and 55 years of age. The life of individuals during this period is developmentally distinct from emerging adulthood in several ways. In contrast to the exploration, information gathering, and career preparation that typically takes place during emerging adulthood, adults in middle adulthood tend to commit to an occupational path and gain greater expertise, take on greater responsibilities, and move up the organisational hierarchy [7]. Additionally, in the family domain, there is a common shift towards commitment and stability that is observed across developed countries, and most developing countries trend in the same direction. Adults in midlife often have children, and a large part of their daily lives is devoted to caring for those children. In some cases, the children would have left the family home or, if still at home, would require less physically demanding care from their parents. These are often the most challenging and demanding years of adult life. Successes or difficulties in handling developmental tasks have the potential to profoundly influence individuals’ current and future lives.

While the patterns outlined seem common in both the family and work domains, demands are likely to vary by ethnicity and socioeconomic background for both emerging and middle-aged adults. Adult development seems, in fact, to be greatly influenced by sociocultural processes [8,9]. Variations in contexts, experiences, life choices, and social roles lead us to become more, not less, heterogeneous as we age [10]. Additionally, changes in life tasks and responsibilities are likely to continue to evolve due to socioeconomic changes as well as family arrangements and gender roles. The resulting intersectional identities are expected to shape adult development, and this process is expected to be sensitive to ingroup or outgroup experiences in countries with racial disparities or social class gaps [11].

Moreover, further support for the role of cultural transition comes from studies on identity development that consider ethnic and racial dimensions (see [12]). According to Williams’ model, for instance, individuals first become aware that ethnic-racial groups are socially meaningful categories with a distinct social image; this is followed by the development of a sense of belongingness, an affective evaluation of the ethnic-racial group(s) of belonging, engagement in performances or role behaviours reflective of cultural values, styles, and language use customs; and, lastly, the attribution of meaning to the behaviours, characteristics, values, and customs that are relevant to one’s ethnic-racial group(s). This process develops through a complex interaction between ecological and individual factors that, depending on the current local context, contribute to the meaning-making process. Accordingly, adulthood is characterised by entry into new roles and environments, understandings of intragroup and intergroup dynamics, and the intersection between several types of identities and the tasks that are expected to be performed.

A comprehensive understanding of the experiences of both emerging and middle-aged adults requires the adoption of a complementary perspective, such as a dynamic systemic framework. It requires an examination of the interplay between multiple factors and forces, individual agency, experiences, encounters with social institutions, and cultural dimensions [13]. This might be particularly meaningful, while simultaneously being complex, for adults who experience cultural transitions. The physical and cultural transition to another country of residence adds complexities to the ‘physiological’ transitions encompassed in the lifespan, exposing individuals to unexpected new demands and risks, such as discrimination [12].

## 2. Adult Immigrants Facing Challenges in Their Transitions

Cultural transitions that are linked to immigration experience lead to several additional challenges for adults besides those already mentioned. The first challenge is identity development: the difficulty immigrants face in re-establishing their career trajectories due to several barriers, including language, unrecognised credentials, and stigmatisation [14]. This can endanger their previous identities at both the professional and social levels, reduce the opportunities for them to be actively included in their host society, and impose new stigmatised identities [14].

The second challenge comes from the labour market. Immigrants’ career opportunities often fall short of their aspirations [15]. As a result, this can lead a significant number of adults with a migration history to unemployment or underemployment, which, in turn, can impact adaptation to the host context and increase the risk of social exclusion [16]. In a study of thirty-one adult refugees of different ages who moved to Germany, immigration was associated with a career break along with the need to cross several contextual barriers to reconstruct one’s career after the resettlement [16]. The three relevant barriers identified in the study are (a) the uncertainty of their residence decision, which makes them feel like they have lost control, making it difficult to adjust their career plans; (b) the lack of personal resources, such as language skills, work credentials, and knowledge of cultural and institutional norms, which reduces their sense of control, restricts their exploration, and results in missed opportunities; (c) the wastage of time, as their previous training and past work Ie would be considered useless. The time needed to learn a new language and sort out the legal requirements to begin looking for work impairs their confidence and sense of control.

A deeper understandinIexperience of navigating diverse cultures and roles underlines the relevance of considering individuals as persons-in-context. This overcomes the intrapersonal influences embodied in the individual, such as personality traits or physical attributes, and reveals dynamic interactions with systems that influence individuals’ experiences at the interpersonal, social, and environmental levels. Assuming that individuals cannot be viewed separately from their societal background when we consider cross-cultural transition, we must focus on the complex process of integration within the host country, which includes acceptance into its labour market [17]. In fact, it is widely recognised that to promote social and cultural inclusion, immigrants need to be supported in constructing their professional lives, because social inclusion largely consists of workforce integration [18,19].

## 3. A Systems Theory Framework (STF) in Understanding Development

The systems thinking perspective provides an overarching framework for many issues in living systems and serves as an integrated framework of human development and behaviour [20]. An STF offers a metatheoretical framework that applies systems thinking to illustrate a complex map of the range of meaningful influences. By considering individuals within their context, the STF maps a range of intrapersonal and contextual influences in diverse, dynamically interconnected systems. These systems are subject to external influence and may also influence other systems beyond their own boundaries. The strengths of systems maps, such as the STF, are their ability to highlight the complex relationships between influences; identify potential interactions between them; determine their nature; understand their impact on problems, choices, and opportunities; and manage and adapt these influences [21].

Systems maps are holistic diagrams that focus on a particular topic [22]. In the case of the proposed STF, the focus is more on work life. Along with systems thinking and systems mapping, the STF has been applied in career counselling, career assessments, and career education for individuals and organisations. With the individual system, social system, and environmental–societal systems that are analysed in the context of time, the STF supports a comprehensive understanding of a person’s identity [23,24]. This has resulted in its application in a range of contexts and cultures [25,26]. However, it can also reveal its potential in understanding and addressing the complex, challenging, and ever-changing personal, social, geographic, and socio-political systems that characterise the migration journey.

### 3.1. A Systems-Theory-Based Perspective on Migration Experience

Recent STF-based research on immigrants and refugees, which was conducted using narrative instruments, identified perceived influences in terms of barriers and opportunities through a meaning-making process that emerged during interviews. Besides systemic barriers, such as language, limited access to local social capital, discrimination, stigma, and lack of recognition of prior qualifications and experiences [27,28,29], recent research underlines the role of other restrictive voices. Through the analysis of narratives of adult immigrants, these were traced as coming from the internal or the proximal context (i.e., family) [19]. Especially for women, familial ties can be destructive, limiting, and silencing. They can serve as a reflection of relational challenges, such as discouragement, oppression, and constraint. In such cases, the social support of the host communities is a source of caring, healthy, and stimulating relationships and new opportunities for autonomy and professional planning. Conversely, immigration may require re-thinking one’s own career projects. Due to the lack of recognition of previous qualifications and experiences, this could mean a reduction of professional ambitions.

In a study conducted by Abkhezr et al. [30] involving three young women from Africa, aged 22–28, three core common themes emerged. (a) ‘Sense of self in transition’ (p. 26), due to the protracted displacement of youth with refugee backgrounds during their migration journey. Furthermore, they had to deal with two transitory processes occurring at the same time: the construction of narratives of self in the maturation process and narratives of self in transition through contexts [30]. (b) ‘Relational resourcefulness for self-reliance in times of tension’ (p. 27); when the participants experienced tensions within communities, such as when they were not aligned with the local cultural values, they searched for support and affirmation in relational sources. (c) ‘The role of storytelling to give voice to participants’ (p. 28), which facilitated the meaning-making process through the active listening of the interviewer. This ‘alloweIpants to experience a relational power dynamic’ (p. 28), activating the reflexivity process on various critical moments, relationships, and social and contextual structures throughout their migration journey.

### 3.2. The Rationale and Aim of the Study

Studies on the cultural transitions of immigrants and their inclusion in the social and professional spheres primarily deal with age by differentiating between minors’ and adults’ experiences [18,31]. To date, little is known about the impact of cultural transition on the roles and tasks that specifically characterise emerging adults in contrast with middle-aged adults.

Our research questions stem from this gap in the literature. This study aimed to explore the systems of influences that characterise emerging and middle-aged adults with a history of migration. More specifically, we explored how factors and agents from the diverse systems in which emerging and middle-aged adults live impact identities, bonds, and experiences of inclusion, as well as their projection into future work life.

Given the complexity of the individual stories of these adults, a qualitative–quantitative study can effectively help identify the dynamic and interacting nature of their personal systems of influences. In the qualitative phase, we identified themes emerging from the theoretically based reflective process activated in the participants and the ones they used to develop their personal system of influences. In the quantitative phase, for each system of influence, we investigated how specific for the two groups of adults were the factors and agents they identified as meaningful influences in their lives. Based on the studies mentioned above, in this first group study, we expected differences between the two groups across diverse systems. Patterns of similarities–differences identified will in turn provide the basis to develop action plans meaningful for the experiences they are going through as adults, effective in addressing perceived barriers and supporting their future personal goals.

## 4. Methods

### 4.1. Research Tool and Procedure

The instrument used to record the interviews is qualitative and derived from the STF of career development. The My System of Career Influences (MSCI) [32] was chosen by the authors to be used for research purposes. This qualitative instrument is a reflexivity booklet that consists of five sections. The tool is directly derived from the STF and the theory on which it is founded. Two of the developers are also the authors of the STF. Additionally, the authors are specialists in career development. The items for the MSCI were developed, tested on different samples, and revised according to the findings of the testing. A pilot version of the instrument involved groups of master’s level students enrolled in career development courses in university settings in Australia and South Africa. Additionally, developmental versions of the MSCI were trialled with a range of career development practitioners [33].

This study focuses on the graphic representations of the diverse maps participants produce and on the visual representations of the systems of career influences. These are found in the first two sections of the booklet.

The first section, titled ‘My present career situation’, consists of open-ended questions relating to the participant’s current career situation, work experience, past and current roles, aspirations, ways of making decisions, and relationships with people who provided them with career advice.

The second section includes a series of diagrams that the participant is encouraged to complete. It indicates perceived factors or agents acting as influences. The first step is the influence system, titled ‘Thinking about who I am’. This system relies on intrapersonal factors (e.g., interests, personality, gender, health, and culture). The second system is titled ‘Thinking about people around me’. In this case, the participant is asked to think about social influences, such as family and friends. The third system, ‘Thinking about society and my environment’ is based on socio-environmental influences, such as the economic system or public transport. The fourth system is titled ‘Thinking about the past, present, and future’. In this system, the user is invited to reflect on their past and future aspirations. The fifth step consists of a diagram titled ‘Representing my system of career influences’, which requires integrating the previous phases into a personal influence diagram. In each system, the participant is invited to identify the most important influences and indicate them with an asterisk. Through a page-by-page reflection, the users are made to identify their most important influencing forces, develop their own personal system of influences, and create an action plan to drive their actions towards reaching their goals.

ThI as”essm’nt was conducted individually in a single session. It was conducted in the Italian language, but both French and English written versions were available to the respondents if required. Basic information in the narratives has been verified by the host communities’ operators.

### 4.2. Participants

The research participants were thirty adults aged 18–54. Some had migrated to Italy from Central Africa (Congo, Zambia, or Cameroon), and most had migrated from West Africa (e.g., Gambia, Ghana, Guinea, Mali, or Nigeria). All participants reported speaking and understanding Italian either well or very well. Their length of residence in Italy ranged from 3 to 5 years for emerging adults and 3 to 15 years for middle-aged adults. The participants lived in the Sicily and Veneto regions.

Based on age, we identified two groups among adults who attended activities and those who were still in contact with local services that support the social and work inclusion of migrants. The first consisted of 15 consecutive participants aged between 18 and 27 years (M = 21.40; SD = 2.72); 12 males and 3 females formed the group of emerging adults. The second group, the middle-aged adults, consisted of an equal number of participants, 12 males and 3 females, aged between 30 and 54 years (M = 39.73; SD = 8.00). The mean education level was 9 years (SD = 3.09) for emerging adults and 10.6 years (SD = 3.04) for middle-aged adults.

About 20% of the emerging adults were enrolled in vocational training or a higher education program, and the other participants were employed in temporary/seasonal or full-time jobs. Regarding marital status, all the emerging adults reported being single except for two participants, and all the middle-aged adults reported being married.

Participants were recruited by the authors through volunteering or convenience sampling. All participants signed an informed consent form prior to joining the research and received written and oral explanations about the study’s aims and procedure. It was also made clear that they could answer as they wished and could choose to not answer questions. When the interviewers believed that the participants were having any difficulty answering, they encouraged them to take their time. Personal, sensitive information, such as the services they referred to for enquiries about work and health issues, was hidden and not considered outside the context of the interviews. The interviewers were instructed to remind the participants that they had the right to terminate their participation in the study at any time.

### 4.3. Data Analysis

The analysis was conducted on all the diagrams, including the final one. Two researchers from the research team conducted the analyses separately. A few discrepancies emerged, which were resolved through the intervention of a third member of the research team. We first identified the central themes that emerged from each diagram. These themes were reported in a matrix, which allowed us to identify their presence or absence.

Subsequently, correspondence analysis (CA) was conducted to evaluate the differences between emerging adults and middle-aged adults. This technique allows us to identify the similarity–difference relationships between the two groups using a contingency table. Correspondence analysis is a widely used technique in research, both quantitative and qualitative, as evidenced by many authors [34,35,36,37]. This technique allows for the improvement of the interpretation and communication of the results of a study [38]. CA applied to qualitative research allows for the measurement of associations (correspondence) between categorical variables (characteristics) and objects, which are coded as 1 or 0: the coding 1 means that a feature was present or mentioned by an interviewee, while 0 corresponds to the absence of features or no mention by the interviewee [39]. The only requirement to consider for applying CA is that the data are coded in such a way that they can be organized into a 4 × 4 cross table [38]. To carry out a correspondence analysis, it is possible to use SPSS [40]. In this study, we used version 25.0 of SPSS.

## 5. Results

Before analysing the results, we evaluated whether the groups were different in relation to their ages. As expected, the analysis showed that there were statistically significant differences between the two groups (*t*_(82)_ = −8.40, *p* < 0.001).

The themes emerging from the reflective activity in each system were then analysed and ordered according to their total frequency of occurrence.

### 5.1. Identifying Intraindividual Determinants

The first step focuses on thinking about the factors that individuals recognise as having a role in defining the factor of ‘who I am’. Participants identified a broad range of influences. The most recurring themes, the characteristics participants primarily used to describe themselves, referred to values, personality, and health. It is interesting to note that Values was the most represented intraindividual category used to define the self (‘*I believe in God and I respect others*’*;* ‘*I did it by facing my responsibilities without a family*’*;* ‘*the value of small helps*’*;* ‘*I like to be a volunteer and I thought I*’*m a person who suffers if others suffer*’*;* ‘*It*’*s crucial for me to be serious, professional, loyal*’*;* ‘*I continued to search, I did not get discouraged*’).

Correspondence analysis showed significant differences in the reflections related to Culture (inertia = 0.16; χ^2^ = 4.82; df = 1, *p* = 0.03). Middle-aged adults provided more classifiable answers in this aspect. Regarding the other choices in the map, no statistically significant differences emerged: Values (inertia = 0.01; χ^2^ = 0.24; df = 1, *p* = 0.62), Personality (inertia = 0.04; χ^2^ = 1.29; df = 1, *p* = 0.26), Health (inertia = 0.11; χ^2^ = 3.39; df = 1, *p* = 0.65), Beliefs (inertia = 0.02; χ^2^ = 0.54; df = 1, *p* = 0.46), Capacity (inertia = 0.01; χ^2^ = 0.14; df = 1, *p* = 0.71), Interests (inertia = 0.02; χ^2^ = 0.56; df = 1, *p* = 0.46), Age (inertia = 0.01; χ^2^ = 0.16; df = 1, *p* = 0.69), Problem-solving strategies (inertia = 0.05; χ^2^ = 1.42; df = 1, *p* = 0.23), Gender (inertia = 0.09; χ^2^ = 2.73; df = 1, *p* = 0.09), Ability (inertia = 0.01; χ^2^ = 0.24; df = 1, *p* = 0.62), and Other (inertia = 0.01; χ^2^ = 0.14; df = 1, *p* = 0.71). The number of choices is reported in Table 1.

Culture represents the core element that impacts self-definition. As evidenced in the collected narratives, emerging adults refer more frequently to aspects of the current context of life *(*‘*I would like to bring some habits in my country and all over Africa*’*;* ‘*here I understood that inclusion is bilateral, it is the creation of a new community of families*’), while middle-aged adults more frequently mention aspects characteristic of their culture of origin (‘*family and reciprocal support was important in the country where I come from*’*;* ‘*there was no need to rush*’*;* ‘*God was very important in my country*’).

Besides culture, it is possible to trace certain trends in the most represented elements that seem to be specific to each age group, despite not being statistically significant. On one hand, personality and interests are more represented in the subgroup of emerging adults (‘*I am the one who then makes the decisions*’*;* ‘*the pleasure of travelling and discovering new things*’*;* ‘*I love music because it makes me feel good*’), while, on the other hand, health, problem-solving strategies, and gender are more frequently chosen in the subgroup of middle-aged adults, with gender generally being mentioned by women describing the difficulties they encountered in determining and reaching their goals.

### 5.2. Representing the Closest Systems of Influence

The second step involves the factor of ‘people around me’: individuals that participants recognise as having a role in defining them and their stories. As in the previous map, the analysis of the responses demonstrates some recurring themes. The most frequent ‘significant others’ identified by the participants as a group were family, friends, and group leaders.

Correspondence analysis demonstrated a significant difference for Family (inertia = 0.22; χ^2^ = 6.65; df = 1, *p* = 0.01). Middle-aged adults provided more easily classifiable answers within the family theme. The other choices did not report statistically significant differences: Friends (inertia = 0.02; χ^2^ = 0.51; df = 1, *p* = 0.46), Group leader (inertia = 0.01; χ^2^ = 0.14; df = 1, *p* = 0.71), Children (inertia = 0.01; χ^2^ = 0.16; df = 1, *p* = 0.69), Manager (inertia = 0.01; χ^2^ = 0.19; df = 1, *p* = 0.67), Reading (inertia = 0.01; χ^2^ = 0.24; df = 1, *p* = 0.62), Newspapers, TV, Internet (inertia = 0.01; χ^2^ = 0.24; df = 1, *p* = 0.62), and Other (inertia = 0.07; χ^2^ = 2.16; df = 1, *p* = 0.14). The number of choices is reported in Table 2.

In contrast to the first system of influences, the patterns are seen to be remarkably similar among the two age groups. The only category that shows differences is Family. Except for two participants, who referred to the family they left in their country of origin (‘*My family in Morocco because there is always from birth to death, she is for you and you for her*’*;* ‘*My family I left because I want to help them*’), participants only refer to the family members they lived with in the new context (‘*I want to work for my family*’*;* ‘*I want to give so many chances to my children*’). Similarly, when participants mention friends and group leaders (frequently people from the work context) they are oriented towards meaningful relationships forged in the new country.

### 5.3. Tracing Influences from Societal and Environmental Systems

The third step, ‘thinking about society and the environment’, further addresses larger systems of influence. As in the previous maps, the analysis of the responses shows some recurring themes, such as the cost of their choices, the areas where they live, the possibility of finding work, and overseas job opportunities.

Correspondence analysis of the third system showed no differences for all choices: Children’s universities/schools (inertia = 0.01; χ^2^ = 0.19; df = 1, *p* = 0.67), Possibility of finding work (inertia = 0.04; χ^2^ = 1.22; df = 1, *p* = 0.27), Cost of choices (inertia = 0.09; χ^2^ = 2.73; df = 1, *p* = 0.10), The area where I live (inertia = 0.01; χ^2^ = 0.14; df = 1, *p* = 0.71), Redevelopment of institutions (same number of choices), Overseas job opportunities (inertia = 0.07; χ^2^ = 2.14; df = 1, *p* = 0.14), Transport (inertia = 0.01; χ^2^ = 0.24; df = 1, *p* = 0.62), and Other (same number of choices). The number of choices is reported in Table 3.

Patterns of influencing factors and agents that characterise the two groups overlap. They trigger reflection about the transition by mentioning the cost of their personal choices (‘*Today you live like this… I understood it on my journey. If I live in Italy, I have to take this style*’) but also identify both the structural characteristics of the context in which they live (‘*I was surprised by the bureaucracy of this country, the documents needed to work*’*)* and the opportunities it provides (‘*how important it was to go to school and meet other people!*’).

### 5.4. Introducing a Time Perspective in the Systems of Influence

When introducing the perspective of time, ‘thinking about my past, present and future’, three main themes emerge from the analysis of the responses collected: work–life balance, meeting a person who is revealed to be of importance, and the openness to move again to find more opportunities.

Correspondence analysis did not show any differences for the following choices: Work–life balance (inertia = 0.02; χ^2^ = 0.60; df = 1, *p* = 0.44), Desire to work overseas (inertia = 0.04; χ^2^ = 1.22; df = 1, *p* = 0.27), Desire not to move (inertia = 0.01; χ^2^ = 0.16; df = 1, *p* = 0.69), Lifestyle I predict (inertia = 0.01; χ^2^ = 0.24; df = 1, *p* = 0.62), I don’t want to work overseas (inertia = 0.07; χ^2^ = 2.14; df = 1, *p* = 0.14), Casual events (inertia = 0.07; χ^2^ = 2.16; df = 1, *p* = 0.14), and Other (inertia = 0.04; χ^2^ = 1.15; df = 1, *p* = 0.28). Significant differences emerged in relation to Meet a person (inertia = 0.16; χ^2^ = 4.82; df = 1, *p* = 0.03). Emerging adults provided more classifiable answers for this choice. The number of choices is reported in Table 4.

Participants sometimes mention a person that influenced their decision to move to Italy (‘*Some people wanted me dead and I had to leave my country’; ‘one day I met a person who was going to Canada and then I started thinking about wanting to leave to find work*’) but, more frequently, they refer to people they met in the new context who supported them or provided helpful suggestions (‘*He gave me a car just because he thought I was a good guy’; ‘thanks to the people who valued my skills in the centre, I’m now a cultural mediator’; ‘I met a young man who sold products for bars and pastry shops. First, I helped him for a while and then I became a supplier too, now we are good friends*’). This is particularly the case for emerging adults.

Besides this specific influence, the pattern characterising the two groups overlaps when introducing a time perspective. The analysis of the narratives collected in this system allows us to identify some patterns in the participants’ stories. First, a break from the past, perceived as threatening and distressing, emerges. This is evidenced by the considerable number of answers oriented towards both present and future times. Here, in turn, there is a predominant focus on job-related meaningful factors, indicating an openness to change, new opportunities, or new life projects within the context of work–life. These are often conveyed through the lens of a meaningful relationship.

### 5.5. Sketching a Personal System of Influences

Finally, the participants filled out maps to demonstrate their ‘individual system of influences’. The most frequent themes observed refer to family, personal characteristics, and values, but there was also a positive orientation towards the future to improve personal life.

Correspondence analysis on selected influencing factors and agents showed no differences in the following choices: Personal characteristics (inertia = 0.02; χ^2^ = 0.56; df = 1, *p* = 0.46), Values (helping others) (inertia = 0.04; χ^2^ = 1.29; df = 1, *p* = 0.26), Interests (inertia = 0.07; χ^2^ = 2.14; df = 1, *p* = 0.14), Family (inertia = 0.01; χ^2^ = 0.14; df = 1, *p* = 0.71), Friends or colleagues (same number of choices), Group leader (inertia = 0.11; χ^2^ = 3.33; df = 1, *p* = 0.07), and Other (same number of choices). Significant differences emerged in relation to the aspect of Improve my life (inertia = 0.13; χ^2^ = 3.97; df = 1, *p* = 0.046). Emerging adults provided more classifiable answers for this choice. The number of choices is reported in Table 5.

Grouping the forces according to the map where participants identified them, both for emerging and middle-aged adults’ intraindividual (14 vs. 11 mentions) and interpersonal systems, namely family and friends (13 vs. 12 mentions), seem to account for the bulk of the forces intentionally selected for participants’ personal system of influences.

This is further confirmed by the answers to the reflective questions proposed after drawing the system of influence. In this last step, participants identified their least important influence as being linked to social image, which is ‘*the others*’ and ‘*the society*’. They also discovered new motivations for themselves during the reflective process activated by the MSCI: ‘*I just found out that I want to try to help the poor*’*,* ‘*How important it was to go to school and meet other people*’*,* ‘*The trust my mother has in me. I had not seriously thought about it before*’.

## 6. Discussion

The literature underlines that the experiences of individuals who navigate adulthood with a personal story of migration are impacted by contextual and systemic variables. For a comprehensive, deeper understanding of the experiences of emerging and middle-aged adults, this study adopted a dynamic and systemic framework. It aimed to give space to the interplay between diverse factors and forces, individual agency, experiences, encounters with social institutions, and cultural dimensions. By engaging in a reflective process, both related and not related to career development, that required participants to share their stories, giving meaning and relevance to the influences they identified and choosing the most significant ones, participants may now play a more active role in constructing their own identities and future stories [41]. The overall results of our study provide fruitful insights into the life trajectories of emerging and middle-aged adults with migration backgrounds.

### 6.1. Systems of Influence in Migrant Adults Navigating Adulthood

A general trend in the number of influences that participants identified can be traced in their systems of influences. In absolute terms, the middle-aged adults involved in this study mentioned a larger number of factors and agents than the emerging adults, possibly because of the greater number of events they could consider during the reflective process.

In line with the available literature, the prevalence of topics related to personality aspects and interests that were considered while reflecting on intraindividual factors suggests that emerging adulthood can be characterised as the age of identity explorations, self-focus, and possibilities [3].

A focus on health emerges, as shown by the consistent number of mentions from middle-aged adults. The issue of health among immigrants has been explored in a large number of studies. These studies highlight the so-called ‘immigrant paradox’, where immigrants tend to have better health outcomes than natives in Western receiving countries, including Italy [42,43,44,45]. At the same time, the length of stay is frequently associated with worsening conditions [46], and in Italy, as in many countries, most of the health-related variables are highly influenced by socio-economic levels, with persisting inequalities in access and use of healthcare due to organisational barriers (namely, language and communication problems, overuse of emergency services, and underuse of primary health care) and structural and interpersonal dynamics, such as biases towards migrants and refugees [47]. The participants in this study have in common a similar length of stay. Issues and health-related concerns are perceived as influential factors mainly by middle-aged adults. The pattern seems more reasonably associated with older age, and the responsibilities and roles that characterise middle-aged adults. But it also shows and underlines a movement from the past to the complexity of their current life, indicating an initial sign of concern and agentic attitude towards their future life.

Also, problem-solving strategies are more frequently identified by middle-aged adults, showing them as being more committed to the expression of their expertise and having greater responsibilities [7]. At the same time, these adults describe themselves as actively addressing the challenges they face from social and work contexts, such as the women who described the difficulties they encountered while setting and reaching their goals.

Besides the differences between the two groups, some commonalities also emerged. While sketching a personal system of influences, the focus was consistently on intraindividual and interpersonal systems, namely family and friends. In line with the literature, feeling connected to others is a key component of well-being for the participants of the study [48]. Additionally, a common contribution to the identity development of both groups revealed by the reflective process comes from the societal and environmental systems, namely the transition and cost of their personal choices connected with the migration journey. However, when considering large systems, their intentional reflective process focuses on the structural characteristics of the context and environment they live in and the opportunities it provides, disregarding the meaningful role played by the people they meet, and any supports or barriers they encounter.

### 6.2. Systemic Influences, Stability, and Exploration in Emerging and Middle-Aged Adults with a Migration Background

Participants in our study demonstrated an interest in moving to other places, thus introducing changes in their activities. This interest in mobility was even more frequent among participants over 30 years of age. This finding opens the door for diverse levels of analysis and lines of interpretation.

The first analysis relies on an intraindividual system of influence. Recent studies suggest that, compared to emerging adults, middle-aged adults experience increased employment rates [11]. This could be indicative of them finding satisfying careers, having limited career change options, or lowering their ambitions. It could also be that middle-aged adults with families face limited career options because of family responsibilities, or they may be unable to take career risks because their income is needed to support their families’ day-to-day needs. Over half of the respondents aged thirty and above in a recent study felt as if they were still in a period full of change, were still discovering themselves as individuals, and were still engaging in career exploration [49]. Some of these adults may find that their career decisions no longer align with their needs or goals. They may face a career and care crunch, during which the competing demands of family and career collide. Alternatively, the cost of supporting a family may necessitate them to seek out higher-paying employment opportunities. Additionally, some may re-evaluate their career path, because this might be one of the last chances to do so.

Finally, we should consider that most participants in the study did not grow up in environments where they perceived themselves as initiative-taking agents in their own career development. Thus, they would be constantly engaged in career exploration to identify aspects of their careers that they like and do not like. Both stability and exploration may thus occur in the career trajectories of migrant middle-aged adults. Their career development may consist of periods of commitment and stability followed by active exploration. Rather than stopping at the end of emerging adulthood, their exploration continued. It is important, then, to investigate how middle-aged adults explore, commit to, and navigate their careers during this period of development, as well as how such exploration and transition are part of an active, ongoing process of personal growth.

A second level of analysis relies on the concept of mobility. If we assume de Haas’ [50] perspective of migration as a freedom, human mobility can be seen as the freedom to choose the place of residence, rather than the actual movement. Moreover, human mobility encompasses the liberty to remain in a location, which is voluntary immobility. This is in contrast with Carling’s [51] notion of involuntary immobility. Additionally, the extent to which migration aspirations persist and materialise in onward migration depends on the degree to which people perceive that their subjective needs and desires can be fulfilled locally. Mobility may represent an individual response to social and economic conditions in the place of residence, regardless of whether this is due to a lack of appropriate opportunities, prevalent inequalities, or threats to personal well-being [52]. However, in general, increased access to innovative ideas through education and the media tends to change people’s ideas about the ‘good life’ in a manner that increases their desire to explore new horizons and move [50].

Following the same line of thinking, recent literature drives attention to the variations in onward migration intentions across generations and the influence of context-related factors or subjective experiences within the new context, rather than considering migration as being related only to individual characteristics [53,54]. For instance, Fokkema and de Haas [54], in their study of groups who immigrated from Africa to southern Europe, demonstrated that immigrants who express the intention to migrate elsewhere are less integrated than those who intend to stay. This was independent of factors such as gender, age, length of stay, and education. Accordingly, Caron’s [53] research highlighted that immigrants’ subjective experiences in their host countries may serve as a relevant dimension to understanding their intention to leave. The migration trajectory, in general, is influenced by opportunities and barriers that vary across different immigrant generations [55]. For instance, the acquisition of EU citizenship plays a crucial role in onward migration. It alters the legal structure of opportunity, and, consequently, immigrants’ ‘motility’ or potential to move [56]. Conversely, migration restrictions tend to encourage long-term settlement, as immigrants are less likely to risk leaving their host country if the prospect of returning is difficult [57]. Thus, the attainment of citizenship can facilitate both onward and return migration. Through the accumulation of ‘migratory knowledge’ [58] or ‘migrant capital’ [59] from previous migration experiences, first-generation immigrants are presumed to have enhanced resources and skills to advance their ‘migratory career’ [55] in another destination country.

### 6.3. Time Perspective and Continuity throughout the Systems of Influence

The concept of time perspective, referring to individuals’ preferred temporal focus on time periods, is under-explored in the literature on immigration [60]. This is further underlined by Boccagni [61] (p. 4): ‘Surprisingly, the future […] is quite marginal from the debate on aspirations and, overall, from migration studies […]’. In his study with immigrant domestic workers, Boccagni [61] found that uncertainty and ambivalence characterised the respondents’ attitudes toward the future and future aspirations, highlighting the need for a more in-depth comprehension of migration-related aspirations, as derived from a combination of micro-, meso-, and macro-factors dealing with time and life-course.

Introducing a time perspective in the reflective process required the participants of our study to reflect on the past, present, and future to identify the most significant sources of influence. Three main themes emerged: meeting a person who had been revealed to be of importance, having a work–life balance, and opening themselves up to future changes to find more opportunities.

A more in-depth analysis of the narratives suggests a discontinuity between the past, which was perceived as menacing and distressing, and present and future life. This was demonstrated by the substantial quantity of responses oriented toward the present and future time periods. Subsequently, an emphasis on the impact of employment is revealed, as evidenced by the sensitivity to work–life balance issues and the desire for new job opportunities.

The stimuli provided by the tool used in the study helped participants develop a dynamic and self-created representation of the reflections from the past, and its sources of influence, to the future, identifying the most helpful elements for planning their future life and career. The overall impression evoked by the data suggests that our participants seem to distance themselves from a past that they experience as painful or incredibly challenging. This is evidenced by the considerable number of answers that were oriented to present and future times. In interpreting this, it is worth noting the absence of explicit mention of determinants from the societal and environmental systems of influences, and the sparse presence of people belonging to the closest systems of influences. In fact, a prevalence of personal characteristics impacts participants’ future planning. Social and environmental dimensions lost their impact when projecting into the future, despite being recognised as influential in their past. Due to the reconstructive nature of the activated reflective process, the selection of the influencing factors and agents is not simply based on a rational process. Rather, it implies holistic meaning-making.

The future that is sketched here is focused on the agentic voice of self [30], which is the most relevant source of proactivity and the most relevant element to rely on when planning their own future. The individuals, rather than the contexts, are seen as the main source of resilience for future challenges rather than the context and persons within it. This seems to be the path they foresee. Although relying on one’s own resources enhances self-determination and individual empowerment, it is not free of risks or obstacles that are difficult to overcome without external support. This seems to indicate a lack of sense of belonging, that is, the experience of personal involvement in a system or environment such that the persons believe themselves to be an integral part of the system or environment [62]. This could also hinder a limited sense of community that, for the immigrant population, may be derived from the experience of settlement in the host country, which encompasses the process of dismantling, accommodating, and reconstructing old and new community contexts [63,64]. Self-perceived belonging may play a significant role since it might reflect integration into social networks and institutions and foster feelings of social solidarity with the core or socially predominant group [65]. As a sense of community facilitates immigrants’ adaptation to new environments, reinforcing their participation in social life and increasing their overall life satisfaction [63], the risk of the participants of our study experiencing isolation, reduced participation in social contexts, and decreased perceived social support appears to be enhanced.

## 7. Conclusions, Limits, and Practical Implications

Our study offers a rich picture of the contextual factors, roles, beliefs, and experiences that influence development in adults with a migration background. While our analyses highlight several common themes as well as systemic determinants in the career development of the adults participating in the study, middle-age in individuals who have migrated must also be understood as a time of increased heterogeneity compared to other life stages [10].

The emerging pattern further underscores the importance of addressing the career aspirations and future construction of all migrants, with specific attention to middle-aged adults, as they are most likely to state their careers as a source of fulfilment in their lives. Paid employment or self-employment can help immigrants experience a dignified life, which supports other processes that facilitate inclusion within the community [66]. This is the reason career support interventions are extremely important in constructing the active participation of immigrants in the host community. As underlined by the OECD [67], there is a need to support immigrants and refugees with quality career guidance and employment support [27]. This will provide them with support to move towards self-sufficiency, better well-being, transitional readjustment, and access to new opportunities. It will also play a role in facilitating the successful transitions of people with immigration backgrounds to meaningful work in their country of resettlement [17,19,27,30].

The methods of this study are not free from limitations. First, the sample consists of thirty participants grouped as emerging and middle-aged adults. This sample size is certainly not representative of all migration histories, especially considering their different countries of origin. Additionally, the results described here refer to adults who are characterised by a specific length of stay in their host country, namely Italy. It is also worth considering the impact of other factors that deal with generational aspects. For instance, it could be worth understanding the meaning of going or not through specific experiences and threats, socio-economic or health-related societal challenges (see the COVID-19 pandemic, among others) when they happen to face responsibilities and tasks typically ascribed to emerging or middle-aged adults.

Moreover, the similarities and differences that characterise the two groups highlight the relevance of considering the specific period of life and its impact on adult development in future studies. Additionally, the systemic perspective conveyed by the MSCI [32] helped the participants and the counsellor reflect on and provide meaning to the context, culture of origin, environment, experience, and other factors that contribute to career construction. As demonstrated in other recent experiences [68], this may result in a process of self-exploration and learning for both clients and career counsellors. The former could have the opportunity to reflect on themselves and their environment to progress in personal development. The latter could help them reflect on social and societal influences excluded from the diagram of personal influences, which, in turn, may reveal future barriers and obstacles to consider in their career development.

## Figures and Tables

**Table 1 behavsci-14-00086-t001:** Thinking about who I am. Number of participants who mentioned influences.

	EA	MA	Total
Values	12	13	25
Personality	11	8	19
Health	6	11	17
Beliefs	6	8	14
Culture *	4	10	14
Capacity	7	6	13
Interests	7	5	12
Age	4	5	9
Problem-solving strategies	3	6	9
Gender	2	6	8
Ability	3	2	5
Other	5	6	11

*Note*. EA = emerging adults; MA = middle-aged adults (* *p* < 0.05).

**Table 2 behavsci-14-00086-t002:** Thinking about the people around me. Number of participants who mentioned influences.

	EA	MA	Total
Family *	5	12	17
Friends	9	7	16
Group leader	5	6	11
Children	4	5	9
Manager	3	4	7
Reading	2	3	5
Newspapers, TV, Internet	2	3	5
Other	1	4	5

*Note*. EA = emerging adults; MA = middle-aged adults (* *p* < 0.01).

**Table 3 behavsci-14-00086-t003:** Thinking about society and the environment. Number of participants who mentioned influences.

	EA	MA	Total
Cost of choices	9	13	22
The area where I live	9	10	19
Possibility of finding work	7	10	17
Overseas job opportunities	5	9	14
Children’s universities/schools	3	4	7
Redevelopment of institutions	3	3	6
Transport	2	3	5
Other	2	2	4

*Note*. EA = emerging adults; MA = middle-aged adults.

**Table 4 behavsci-14-00086-t004:** Thinking about my past, present, and future. Number of participants who mentioned influences.

	EA	MA	Total
Work–life balance	11	9	20
Meet a person *	10	4	14
Desire to work overseas	5	8	13
Desire not to move	4	5	9
Lifestyle I predict	2	3	5
I do not want to work overseas	2	0	2
Casual event	1	4	5
Other	3	1	4

*Note*. EA = emerging adults; MA = middle-aged adults (* *p* < 0.05).

**Table 5 behavsci-14-00086-t005:** My individual system of influences. Number of participants who mentioned influences.

System		EA	MA	Total
Who I am	Personal characteristics	5	7	12
	Values (helping others)	7	4	11
	Interests	2	0	2
People around me	Family	9	8	17
	Friends or colleagues	4	4	8
	Group leader	0	3	3
	Improve my life *	7	2	11
	Other	3	3	6

*Note*. EA = emerging adults; MA = middle-aged adults (* *p* < 0.05).

## Data Availability

The data are unavailable due to privacy or ethical restrictions.

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
