# Peer review of "Navigating Adult Life from Emerging to Middle Adulthood: Patterns of Systemic Influences and Time Perspective in Migrants"

_behavsci, 2024, doi:10.3390/bs14020086_

Round 1

Reviewer 1 Report

Comments and Suggestions for Authors Regarding the data collection instrument used, its psychometric qualities are not made known, with which, we start from the use of a data collection instrument, on which the entire investigation is based, of which it is not known if it evaluates what it claims to evaluate, if it is accurate and if that instrument has systematically obtained the same results in similar situations. Regarding the participants, there is no information about the population to which the research is directed and, also, there is no information about the type of sampling used. The use of the SPSS package to analyze qualitative data is not common, above all, the results on which we are going to work do not provide qualitative information, which, in itself, is incongruent.
Therefore, I consider it important that each of these methodological elements be reviewed, in addition to placing greater emphasis on the type of research, methodology and research design, since it is necessary information to understand the research and is not collected.

Author Response

On behalf of our writing team, I would like to express our gratitude for the time devoted to our manuscript. We are appreciative of the thoughtful feedback provided by the reviewers. We have taken these suggestions into consideration while working our manuscript and are hopeful the edits reflect the comments and feedback of the reviewers.  The changes we made  are highlighted for your reference. Again, we really appreciate your help to make this manuscript stronger and more detailed.

Reviewer 2 Report

Comments and Suggestions for Authors

The manusctipt under review presents the results of using a systems theory framework to the experiences of adult migrants experiencing cultural transitions. The results enhance understanding of the lives of emerging and middle adults with such backgrounds, and the authors discuss the practical implications of their findings.

Overall, the manuscript is strong throughout. Well-written, well-sourced, well-organized and with clearly presented methodology and results. The  reporting of results should be corrected such that statistical symbols are italicized. I would also recommend using italicized symbols for chi-square instead of spelling it out.

Good work to those involved.

Author Response

On behalf of our writing team, I would like to express our gratitude for the time devoted to our manuscript. We are appreciative of the thoughtful feedback provided by the reviewers.   Both in the cover letter and in the manuscript have the changes we made are highlighted

Reviewer 3 Report

Comments and Suggestions for Authors

First of all, I want to thank the authors for the interesting article on a very important topic. I have read it with pleasure. However, there are some points to which I would like to draw the authors’ attention. It should be stated and made clear that my competences are not the methodological aspects, but rather the field of migration and issues concerning labour market integration.

The authors want to explore how socioeconomic factors and ethnicity are part of a systemic context that the migrants are asked to consider when mapping and interpreting meaningful influences. It could be enhanced how these affect development tied to becoming an adult and onwards in the middle adulthood. The article would gain from earlier introducing some of the content in the first section of the discussion. For me, that is when some important points the authors’ want to make fell into place.

In my reading of the results and the article, it seems that age is still the paramount factor which is considered, with interesting results such as that the understudied group of middle adulthood continue exploration and affecting behaviour such as patterns of mobility depending on how they understand their contexts and influences. It needs to be made clear how socioeconomic factors actually gain importance. One example is health, which seems to be a relatively important factor, mentioned by 17 respondents. In the discussion (l. 400), though, it gets hardly any attention at all. It is hard to understand why this is left so uncommented when resilience is another point made. For example, mental health seems to be quite interesting in relation to the (strategic?) focus on an agentic self and the shutting out of the past, which it is stated is talked about by very few.

This is also connected to a comment I have concerning migrants behaviour on the labour market and how they “often fall short of their aspirations” (l. 75). This wording can be nuanced. Many, of course rightfully, aspire to find an apt employment in relation to their education/training/experience. Other migrants, however, are very pleased with any employment at all, given the mechanisms of labour markets (which can certainly strive for flexibility in the job seekers and have need for people who will take any position) this has been known to create lock-in effects that make it hard to find a suitable position. Such phenomena are also considered further down in the article (l.134-135; l. 427).

I think the article would gain perhaps from also considering generational aspects. Focus or not focus on family, aspects concerning religion and so on may also be the result of generational discourses regarding how they are viewed. Thus, it would not only be a matter of age and developmental stages, but also a time you grew up and were formed in. That may, however, be a different study altogether.

There is also a couple of minor comments, that I have. Firstly, I think the sentence “no reflections have been raised on cultural transition experiences or their impact on the roles and tasks that emerging and middle adults have to face” (l. 152-153), is a bit general and strong. Maybe it is because it could be further described in which field this has not been considered. I can think of some studies that have explored that but maybe I am missing something.

Secondly, although I am not that familiar with the methodology and tools used in this study, I find it a bit hard to read the tables. There seems no hierarchy or particular order in which the different factors are presented. Perhaps that is not necessary, but it made me wonder and start doing that work myself, as a reader. The text before and after, in the sections where results are presented, also give few clues to the order in which they are discussed. This could be clarified.

Author Response

(The authors gave the same response as above.)

Reviewer 4 Report

Comments and Suggestions for Authors

Comments for the Authors

The article “Navigating adults’ life from emerging to middle adulthood: Patterns of systemic influences and time perspective in migrants” deals with an important and contemporary subject and offers some new insights into the migration field of research and the analysis of migration drivers and determinants. Especially significant is the use of the systems theory framework and My System of Career Influences instrument to explore the systems of influences that characterise emerging and middle adults’ identities, bonds, and experiences of inclusion, as well as their projection into future work life.

The structure of an article corresponds to the main propositions of academic writing, the text is clear and unambiguous, and conclusions and implications are sound and derived from the obtained results. Discussion brings especially insightful and elaborative interpretation of obtained results which could be used beyond scientific research.

Even though it was not conducted on a representative sample, the proposed method and used instruments could be tested in other contexts. The analytical procedure and results are clearly presented.

There are several minor remarks for authors to consider:

  1. I would suggest including the correspondence analysis results in the presented tables to be able to follow the numbers more easily and to make the text more fluent.
  2. Though the name of the instrument used includes the term “Influence” I would suggest substituting it in the text with the term effects i.e. affecting. In qualitative research, there is no control and experimental group so it would be hard to prove exact influences: see for instance l. 56 or l. 536.

Author Response

On behalf of our writing team, I would like to express our gratitude for the time devoted to our manuscript. We are appreciative of the thoughtful feedback provided by the reviewers. We have taken these suggestions into consideration while working our manuscript and are hopeful the edits reflect the comments and feedback of the reviewers.  The changes we made are  highlighted for your reference. Again, we really appreciate your help to make this manuscript stronger and more detailed.

4.1. To follow the numbers more easily and to make the text more fluent, we have ordered the tables based on the total choices of both groups and inserted, for each table, the significant differences (indicated with an asterisk). (see all tables)

4.2 The word “influence” has been maintained when it explicitly refers to the framework and the tool. In occurrences such as the ones mentioned by the reviewer, and according to the specific context, the word “influence” has been substituted with terms such as influencing factors, agents, determinants, impacts. 

Round 2

Reviewer 1 Report

Comments and Suggestions for Authors Once the proposed changes have been made, the article should be accepted in its current form

Reviewer 3 Report

Comments and Suggestions for Authors

Well done!